# Native Smartphone Single- and Dual-Frequency GNSS-PPP/IMU Solution in Real-World Driving Scenarios

**Ding Yi, Sihan Yang * and Sunil Bisnath**

Department of Earth and Space Science and Engineering, York University, Toronto, ON M3J 1P3, Canada;
dingyi@yorku.ca (D.Y.); sbisnath@yorku.ca (S.B.)
* Correspondence: syang551@yorku.ca

**Abstract:** The Global Navigation Satellite System (GNSS) capability in smartphones has seen significant upgrades over the years. The latest ultra-low-cost GNSS receivers are capable of carrier-phase tracking and multi-constellation, dual-frequency signal reception. However, due to the limitations of these ultra-low-cost receivers and antennas, smartphone GNSS position solutions suffer significantly from urban multipath, poor signal reception, and signal blockage. This paper presents a novel sensor fusion technique using Precise Point Positioning (PPP) and the inertial sensors in smartphones, combined with a single- and dual-frequency (SFDF) optimisation scheme for smartphones. The smartphone is field-tested while attached to a vehicle's dashboard and is driven in multiple real-world situations. A total of five vehicle experiments were conducted and the solutions show that SFDF-PPP outperforms single-frequency PPP (SF-PPP) and dual-frequency PPP (DF-PPP). Solutions can be further improved by integrating with native smartphone IMU measurements and provide consistent horizontal positioning accuracy of <2 m rms through a variety obstructions. These results show a significant improvement from the existing literature using similar hardware in challenging environments. Future work will improve optimising inertial sensor calibration and integrate additional sensors.

**Keywords:** PPP; smartphones; IMU; single- and dual-frequency combination; ionospheric constraints; GNSS outage

## 1. Introduction

Smartphone positioning techniques and their navigation capability based on Global Navigation Satellite Systems (GNSS) have seen intensive development in the last decade. The proliferation of GNSS-enabled smartphones and wearables has boosted the evolving industry of Location-Based Services (LBS), around which a growing number of applications such as lane-level navigation, personnel/property monitoring, augmented reality, etc., are thriving. Many of these applications require greater user positioning accuracy, resilience, and/or availability.

It has been a topic for both industry the research communities to innovate within the low-cost domain to address these challenges and improve solutions further. The idea for improving smartphone GNSS positioning is often two-fold: (1) improving GNSS processing techniques to increase solution accuracy; (2) using sensor fusion to enhance accuracy, and more importantly, resilience and availability. Compared to geodetic-grade hardware that is traditionally used in the surveying and mapping community, ultra-low-cost GNSS modules suffer from poor signal reception, low gain, or poor multipath suppression [1–3]. On improving GNSS-only solutions, progress has been made in evaluating smartphone signal strength and carrier to noise ratios [4–7], observation noise characteristics and optimisation [8,9], duty cycling [10,11], as well as precise positioning techniques and their further enhancements [12–14].

In 2016, Google announced the availability of code and phase raw measurements to smartphone users [15], enabling more precise positioning techniques such as Precise

Point Positioning (PPP), which is capable of delivering centimetre-level positioning in minutes with a standalone geodetic receiver and precise products [16,17], without the base station and baseline length constraints [18,19]. Most of the earliest smartphone positioning performance assessments focus on single-frequency (SF) processing, ref. [20] conducted a single-frequency PPP static experiments, and obtained horizontal and vertical rms of 37 cm and 51 cm, respectively, under the open-sky environments. Similar conclusions were also drawn by [21]. Following these studies, ref. [22] utilized the ionospheric-constrained, single-frequency PPP strategy to further improve smartphone positioning, and results demonstrated that the level of sub-metre accuracy can be reached with the Mate 30 smartphone in static tests.

Thanks to the emergence of multi-constellation, dual-frequency smartphone GNSS chipsets, it is now possible to utilize more observations and manage the ionospheric delays by means of, e.g., ionospheric-free combination or ionospheric error estimation. In 2018, the first dual-frequency (DF) smartphone MI 8 was released with a BCM47755 chip [23] and in this context, a host of studies demonstrated that the ionospheric-free dual-frequency MI 8 PPP solutions may achieve decimetre-level accuracy in static environments with real-time [24] or final products [25]. Continuing this research with dual-frequency processing, ref. [26] comprehensively compared the PPP performance with four released smartphones and an average horizontal error of 40 cm can be obtained for dual-frequency MI 8 solutions, which was superior to single-frequency solutions, but the performance degraded to 6 m in a kinematic test. Recently, a subsequent contribution from [27] demonstrated that ionospheric-constrained dual-frequency PPP is able to benefit smartphone positioning significantly compared to low-cost and geodetic-receivers, in particular in suburban environments. Meanwhile, ref. [28] conducted a walking experiment and achieve 0.85 m and 1.09 m in horizontal and vertical components, respectively, with the aid of the real-time ionospheric products.

It is often necessary for PPP or other GNSS techniques to fuse solutions with measurements from additional sensors to maintain a similar level of accuracy during GNSS outages. For smartphone navigation, there is a stronger need due to poor GNSS measurement quality, as well as a tendency from users to acquire positioning solutions in obstructed environments such as urban canyons. Traditionally, it has been shown that during GNSS outages, low-cost MEMS (Micro-ElectroMechanical System) IMUs have the potential to achieve decimetre-level accuracy over 60 s when coupled with geodetic-grade GNSS receivers in PPP mode [29], or at the metre-level for a few seconds with low-cost, single-frequency GNSS receivers [30]. Fusing more recently available low-cost, dual-frequency GNSS receivers in PPP with MEMS-IMU produces decimetre-to-metre-level accuracy during 30 s of outages with four visible satellites, and dual-frequency processing gives a significant edge of 10 times improvement over single-frequency processing [31]. The emergence of the latest smartphone-grade, or ultra-low-cost GNSS chipsets has led to studies investigating a native sensor fusion scheme using the onboard inertial sensors of smartphones [32–34]. However, in smartphone PPP processing, there is little literature which investigates GNSS-PPP/IMU fusion specific to the ultra-low-cost GNSS receivers and IMUs. An earlier study by the authors, has shown the potential of bridging solution gaps produced from GNSS outages, while maintaining a metre-level solution [35] using smartphones strapped on top of vehicles. It has yet to been seen if a feasible GNSS-PPP/IMU processing scheme that brings the performance to a similar level in real-world driving environments where complex obstruction and multipath profiles are involved.

In spite of this remarkable progress, the major limitations restricting the use of smartphones for precise navigation applications are their low-quality noisy measurements and positioning degradation during GNSS outages. Thus, utilizing single- and dual-frequency (SFDF) observations, as well as IMU information are vital for smartphone navigation. Ref. [36] proposed to use the single-frequency ionosphere-corrected code measurements with dual-frequency ionospheric-free code and phase measurements for low-cost GNSS device position determination, and in this context, ref. [37] proved that this approach

would benefit smartphone positioning through walking experiments. While single- and dual-frequency PPP is not a novel concept and there are some studies focusing on SFDF scheme for low-cost and smartphone devices, it should be noted that this paper uniquely explores the benefits of SFDF strategy for smartphone not only in benign environments, but also in realistic (automotive) suburban areas where smartphone GNSS signals tend to be blocked or affected by multipath effects. For this research, this paper also employed the native inertial sensor from smartphones and developed a single- and dual-frequency PPP engine enhanced with ionospheric constraints (PPP-IC). Therefore, the main significant novelties and contributions of this work aim to answer the following research questions:

1.  How does single- and dual-frequency PPP processing improve smartphone GNSS positioning performance and how does it compare with other PPP processing strategies (single-frequency PPP and dual-frequency PPP) in GNSS challenged environments?
2.  How does smartphone IMU dead-reckoning perform compared to other low-cost MEMS IMU? How does the inclusion of the smartphone inertial sensor affect PPP solutions?
3.  What is the "best" positioning performance that smartphones can achieve with multi-GNSS PPP/IMU integration in real-world driving environments?

## 2. Mathematical Models and Data Processing Strategies

This section introduces the theoretical background behind single- and dual-frequency PPP-IC and IMU tightly-coupled strategy, as well as the York-PPP user processing engine parameter settings.

### 2.1. Single-Frequency and Dual-Frequency PPP-IC Model

Ionospheric delay is a critical error source for PPP processing. The traditional ionospheric-free (IF) model is capable of eliminating the first-order ionospheric delay through the combination of dual-frequency observations [38,39]. Unlike the IF PPP, uncombined PPP is able to use external ionospheric delay information to benefit positioning performance. Recently, refs. [40,41] rewrote the uncombined PPP observation equation by decoupling the receiver DCB with estimated slant ionospheric delay. In this context, ref. [27] suggests that ionospheric constraints (IC) bring significant benefits to low-cost devices, and this PPP-IC equation can be written as:

$$P_{r,1}^s = \rho_{r,1}^s + (cdt_r + b_{r,P_{IF}}) + T_r^s + I_{r,1}^{s,constrained} - \frac{f_2^2}{f_1^2 - f_2^2}DCB_r + \varepsilon_{P_1}$$

$$\Phi_{r,1}^s = \rho_{r,1}^s + (cdt_r + b_{r,P_{IF}}) + T_r^s - I_{r,1}^{s,constrained} + \frac{f_2^2}{f_1^2 - f_2^2}DCB_r + \tilde{N}_{r,1}^s + \varepsilon_{L_1}$$

$$P_{r,2}^s = \rho_{r,2}^s + (cdt_r + b_{r,P_{IF}}) + T_r^s + \frac{f_1^2}{f_2^2}I_{r,1}^{s,constrained} - \frac{f_1^2}{f_1^2 - f_2^2}DCB_r + \varepsilon_{P_i}$$

$$\Phi_{r,2}^s = \rho_{r,2}^s + (cdt_r + b_{r,P_{IF}}) + T_r^s - \frac{f_1^2}{f_2^2}I_{r,1}^{s,constrained} + \frac{f_1^2}{f_1^2 - f_2^2}DCB_r + \tilde{N}_{r,2}^s + \varepsilon_{L_2} \qquad (1)$$

$$\tilde{N}_{r,1}^s = N_{r,1}^s + b_{r,\Phi_1} - b_{\Phi_1}^s - b_{r,P_{IF}} + b_{P_{IF}}^s - \frac{f_2^2}{f_1^2 - f_2^2}DCB_r$$

$$\tilde{N}_{r,2}^s = N_{r,2}^s + b_{r,\Phi_2} - b_{\Phi_2}^s - b_{r,P_{IF}} + b_{P_{IF}}^s - \frac{f_1^2}{f_1^2 - f_2^2}DCB_r$$

where $P_{r,i}^s$ and $\Phi_{r,i}^s$ are the pseudorange and carrier-phase measurements on frequency $i$ ($i \in \{1, 2\}$); $\rho_r^s$ is the geometric distance between satellite $s$ and GNSS receiver $r$; $c$ is speed-of-light in vacuum, and $dt_r$ refer to receiver clock offsets; $b_{r,P_{IF}}$ and $b_{P_{IF}}^s$ represents the receiver and satellite code biases in IF combination, respectively; likewise, $b_{r,\Phi}$ and $b_\Phi^s$ represent, respectively, the receiver and satellite phase biases; $T_r^s$ is the slant troposphere delay; $f$ refers to the signal frequency, and $I_{r,1}^{s,constrained}$ is estimated ionospheric delay

constrained by external GIM (Global Ionosphere Map) information on first frequency. $\tilde{N}_{r,i}$ is the estimated carrier-phase ambiguity, containing code and phase biases; $DCB_r$ is the estimated receiver DCB; $\varepsilon_P$ and $\varepsilon_\Phi$ are, respectively, pseudorange and carrier-phase unmodelled errors including measurement noise and multipath error.

*2.2. Tightly-Coupled PPP/IMU Model*

The inertial sensor serves as the dead-reckoning sensor that provides relative displacement information that determines the user's position based on the previous position. Rather than absolute, position-fixing measurements such as GNSS, an IMU requires initialization from known knowledge to compute a navigation solution. GNSS solution suffers from the inherent disadvantage of a space-based ranging system where challenging environments degrade or disable position solutions. Fusing the measurements from the native IMU within smartphones provides (1) a dead-reckoning solution from mechanization equations and (2) an improved EKF (Extended Kalman Filter) solution that employs sensor fusion. The section briefly reviews the key ideas in mechanization and EKF fusion with PPP.

Typically, in a GNSS/IMU fused system, the IMU has a higher data rate than GNSS receivers. In the case of smartphones, typically the GNSS sensor has a one-second sampling interval and a varying data frequency from IMUs due to internal power-saving or other factors within the operating system. To unify IMU input, the IMU data are pre-processed through interpolation to produce a uniform 100 Hz data stream [35].

Mechanization enlists a series of deterministic physical equations to compute the updated attitude, velocity, and position based on known information and current IMU accelerometer/gyroscope readings. This solution, without GNSS measurements is considered the IMU-only solution in the subsequent analysis, or the dead-reckoning solution. The IMU measures specific force $f_{ib}^b$ and angular rate $\omega_{ib}^b$ as inputs to the mechanization equations. Here, the subscript $b$ refers to the IMU body frame, and $i$ refers to the inertial frame. In this context, the specific force measured $f_{ib}^b$ should be interpreted as measurements in the body frame with respect to the inertial frame, resolved in the body frame axes. The inertial solution coasted from the last GNSS observation is carried onto the next available GNSS epoch for the EKF.

A tightly-coupled integration approach is used in this study. In tightly-coupled GNSS/IMU integration, the EKF takes in GNSS measurements and updates both IMU and GNSS states in a centralized approach. The total-state vector is defined as Equation (2) [31,42,43]:

$$\delta X = [X_{x,y,z} \quad V_{x,y,z} \quad \epsilon_{x,y,z} \quad \delta I_r^s \quad \delta T_r^s \quad \delta t \quad \delta \dot{t} \quad b_{ax,ay,az} \quad b_{gx,gy,gz} \quad N^s] \tag{2}$$

where $X_{x,y,z}$ denotes the position states; $V_{x,y,z}$ denotes the velocity states; $\epsilon_{x,y,z}$ denotes the attitude in local navigation frame; $\delta I_r^s$ and $\delta T_r^s$ denote the slant ionospheric and tropospheric delay, respectively; $\delta t$ denotes receiver clock errors; $\delta \dot{t}$ denotes receiver clock drift; $b_{ax,ay,az}$ and $b_{gx,gy,gz}$ denote bias in accelerometer and gyroscope, respectively; and $N^s$ denotes the float ambiguity terms.

The design of the PPP/TC EKF is based on a conventional closed-loop model demonstrated in Figure 1. The raw measurements conditioned through pre-processing are fed into the mechanization equations to produce an IMU-only solution. Once the next GNSS observation set becomes available, the GNSS-PPP/IMU EKF runs to produce a solution. Finally, the bias estimates for accelerometers and gyroscopes are fed back to the IMU data reading in the closed-loop feedback approach for the next epoch. The feedback is additive to raw $f_{ib}^b$ and $\omega_{ib}^b$ inputs. Since mechanization runs using the state vector inherited from the previous epoch, an accurate dead-reckoning solution requires an accurate fused prior epoch estimate. The EKF employs zero-velocity update (ZUPT) to improve solution quality when the vehicle is stationary.

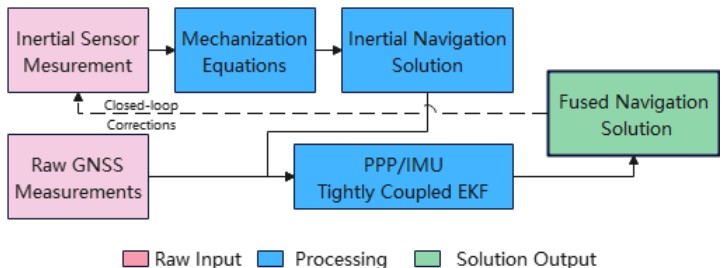

**Figure 1.** Schematic of the tightly-coupled Extended Kalman Filter in York-PPP user processing engine.

### 2.3. York-PPP Engine and Processing Settings

The York-PPP engine is a well-established software that is able to produce PPP/IMU tightly-coupled solutions for geodetic, low-cost, as well as smartphone devices. Table 1 highlights relevant engine settings for this current processing with corresponding products.

**Table 1.** York-PPP setting and corresponding strategies.

| York-PPP Settings | Products/Value |
|---|---|
| Satellite orbit and clock | GFZ rapid products (GBM) [44] |
| Tropospheric delays | Hydrostatic: GMF model [45]<br>Wet: estimated |
| Ionospheric delays | Constrained by GIM (final IGSG products) [46] |
| Satellite DCB | CAS products [47] |
| Weighting scheme | Carrier-to-noise ($C/N_0$) based |
| Elevation mask | 10° |

All smartphone collected raw measurements were processed in the aforementioned PPP-IC mode with and without IMU integration. Besides correcting the satellite orbits and clocks with GFZ (Geo-ForschungsZentrum) rapid products, the satellite DCBs are corrected using CAS (Chinese Academy of Sciences) products. The combined final IGSG GIM products served as the external constraints for ionospheric error estimation. Meanwhile, a carrier-to-noise ($C/N_0$) based stochastic weighting scheme was adopted since the smartphone signals contaminated with significant multipath errors tend to have lower $C/N_0$ ratios [5], and corresponding standard deviation of the code and phase observation $\sigma$ can be estimated as Equation (3) [48]:

$$\sigma = a + b * 10^{-\frac{1}{2} * \frac{C/N_0}{10}} \tag{3}$$

where coefficient $a$ is 4 m and 6 cm for pseudorange and carrier-phase observations, respectively. These values are empirically derived from the residuals. In addition, $b$ is the pseudorange chipping length and carrier-phase wavelength for pseudorange and carrier-phase measurements, respectively [7].

In terms of measurement quality control, the satellite elevation cutoff angle is chosen as 10°and, on average, 0.7 satellites are rejected per epoch for four constellations. This selection ensures that low-elevation satellites will be removed owing to their high noise and multipath. Additionally, the PPP engine is automatically configured to screen out satellites with post-fit residuals tenfold larger than the standard deviations of measurements to mitigate the impacts of outliers.

### 3. Measurement Campaigns

Real-world experiments were designed to imitate daily driving scenarios where the user places their phone on the vehicle's dashboard. This section elaborates on the exper-

imental procedures used to provide background material for explaining and analyzing positioning results.

### 3.1. Equipment Setup

The experimental setup contains two rover sensor collections: (1) the reference sensors and (2) the smartphones, as well as an RTK base station as illustrated in Figure 2.

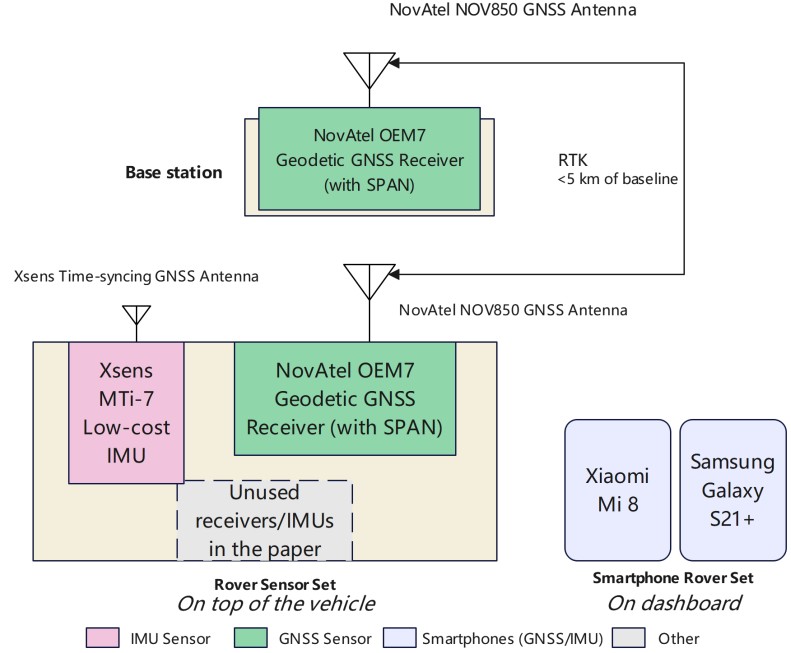

**Figure 2.** Schematic of equipment setup for measurement Campaigns.

The hardware used are a collection of geodetic and automotive-grade GNSS and IMU sensors to produce a reference positioning solution and raw measurement comparisons. The sensors are mounted on and in a box on top of the vehicle as shown in Figure 3. A NovAtel SPAN geodetic receiver + IMU combination and geodetic antenna is utilized, and a NovAtel base station is located on an open rooftop within a 5 km baseline to generate a post-processed, smoothed RTK/IMU tightly-coupled reference solution using the Inertial Explorer software. An Xsens MTi-7 automotive-grade IMU is also used to provide a low-cost inertial sensor alternative to the smartphone IMUs.

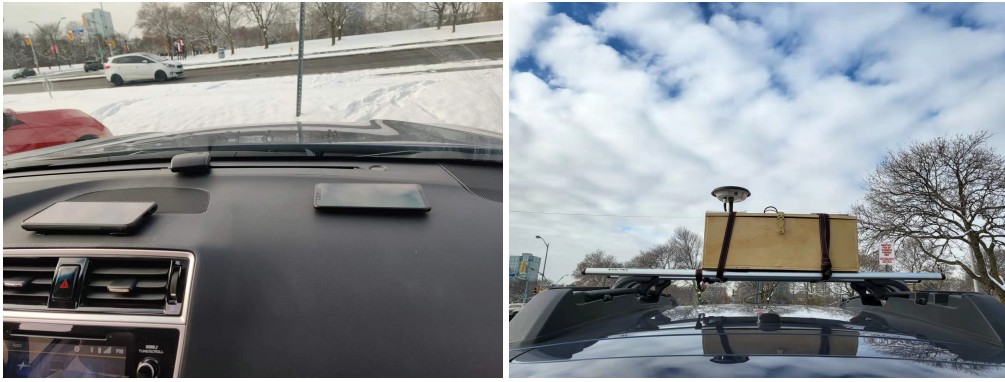

**Figure 3.** Photos of strapped reference sensors used in measurement campaigns.

Two phone models are used in this study, the Xiaomi Mi 8 and Samsung Galaxy S21+. Both support multi-constellation (GPS+GLONASS+Galileo+BeiDou) tracking, as well as dual-frequency signals (L1 and L5). The GnssLogger (Google) and Geo++ Rinex Logger

(Geo++) apps are used concurrently to log raw IMU measurements and GNSS observations, respectively. To mimic real-world driving, the phones are attached on the dashboard instead of placing on the vehicle roof. This study assumes the IMU frame to remain stationary relative to the vehicle frame throughout the experiment. The smartphones are equipped with consumer-grade inertial sensors. Their factory specifications are detailed in Table 2 in comparison with the automotive/industrial-grade Xsens MTi-7 example used alongside. The former class of inertial sensors usually cost less than 10 USD, whereas the latter class can cost up to 1000 USD. The specifications show a general weaker performance of smartphone IMUs. Traditionally, it is expected that GNSS-denied navigation outages using an automotive/industrial grade IMU will be less than one minute. By extension, smartphone IMUs should fill shorter GNSS-denied navigation outage periods. For the IMU model in Galaxy S21+, in-run bias is not found in factory datasheet or in the literature, and future work can include static IMU testing to compute Allan variance.

**Table 2.** Factory specifications of the inertial sensors used in experiment [34,42,49–51].

| Phone Models | Xiaomi MI 8 | Samsung Galaxy S21+ | Xsens MTi-7 |
|---|---|---|---|
| **IMU Model** | InvenSense ICM-20690 | ST-Microelctronics LSM6DSO | N/A |
| **Gyroscope** | | | |
| **In-run bias stability (°/h)** | >1000 | - | 10 |
| **Noise density (°/s/$\sqrt{Hz}$)** | 0.004 | 0.004 | 0.003 |
| **Standard full range ($\pm$°/s )** | 2000 | 2000 | 2000 |
| **Accelerometer** | | | |
| **In-run bias stability (mg)** | 40 | - | 0.03 |
| **Noise density (µg/$\sqrt{Hz}$)** | 100 | 110 | 70 |
| **Standard full range ($\pm$g)** | 16 | 16 | 16 |

### 3.2. Environmental Consideration

Since this study aims to mimic daily driving, a route including open parking lots with vegetation, suburban roads and mixed underpasses was chosen. Examples of Google Street View photos is shown in Figure 4. Road tests were performed on three separate days using an identical route around York University, Toronto, Canada, and a total five datasets from the two phones were collected. The route takes 26, 16, and 24 min for each road tests over a total driving distance of approximately 10 km.

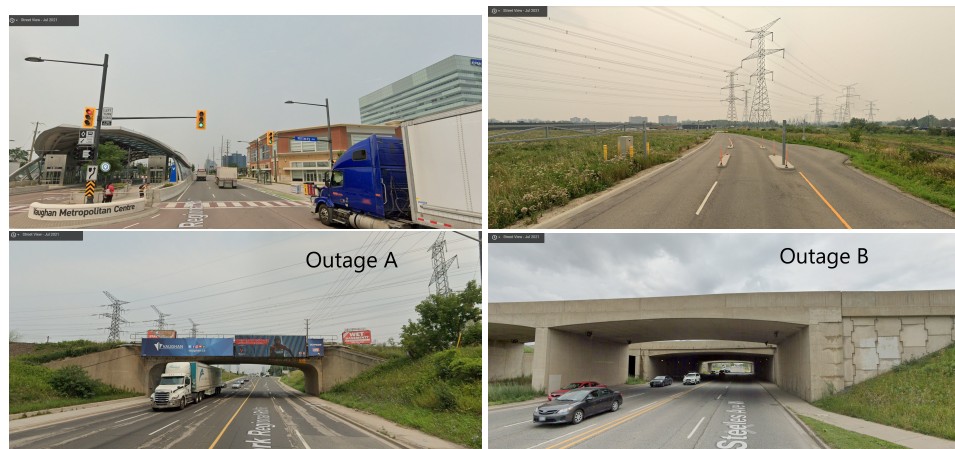

**Figure 4.** Google Street View of experimental environments.

Based on the example collected on 30 November 2021, the effects of outages are briefly discussed with a more detailed quantitative analysis to follow. Outage A is caused by a railway bridge overpass with a width of approximately 6 m over an arterial road. The overpass caused a reduction in the number of visible satellites for over 3 s, approximately 5 min from the start of data collection. Outage B is a more complicated underpass below a major highway and two on/off ramps, consisting of three separate complete outages over 100 m with two small gaps, each approximately 10 m. The duration of Outage B is ∼10 s at an elapsed time of 15 min from the beginning of the drive. The complexity increases the difficulty in producing a good GNSS solution as momentary GNSS signal re-acquisition produce poor quality measurements and the filter has little time to converge before subsequent outages.

## 4. Results Analysis

In Section 4.1, the dead-reckoning performance behaviour is first demonstrated, followed in Section 4.2 by the assessment and analysis of proposed processing strategies. The findings are then further summarised with statistics presented in Section 4.3. Finally, in Section 4.4, single- and dual-frequency PPP/IMU solutions under different scenarios are investigated.

### 4.1. IMU Dead Reckoning Performance

An extreme, but realistically meaningful scenario is complete GNSS outages such as driving in tunnels. This study first attempts to produce IMU-only solutions over two separate 60 s GNSS outages where the positioning solution is generated solely from IMU inputs. To simulate such outages, outages A and B are extended by not including GNSS measurements during the 60 s. The solution states are first initialized using a reference solution produced by SPAN, with lever arm correction to the smartphone frame. The horizontal errors are compared against the SPAN reference solution in post-processed RTK/TC with full access to all available measurements in Figure 5.

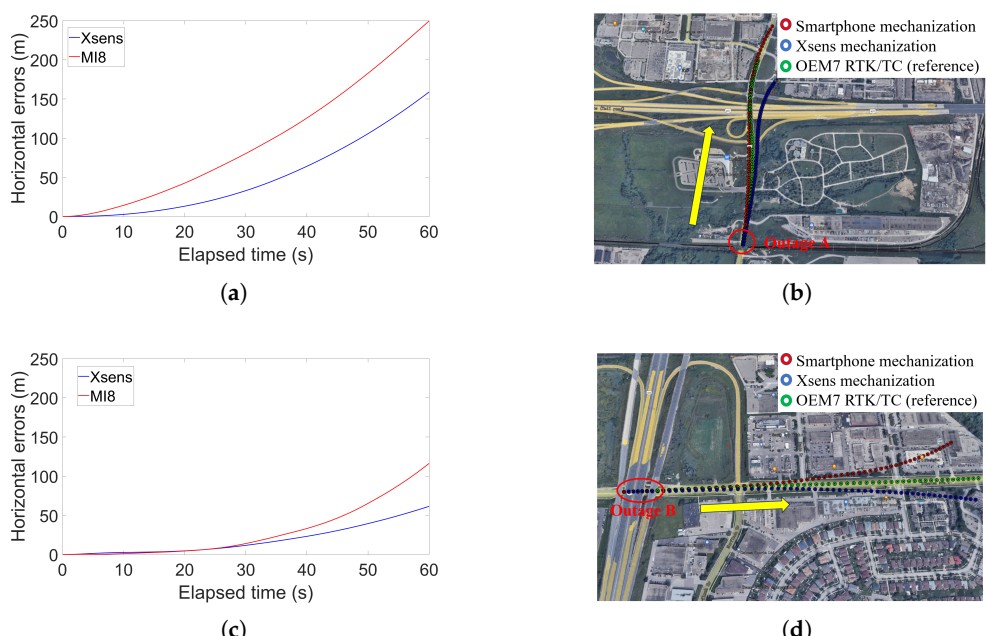

**Figure 5.** Position estimates based on dead-reckoning of Xsens and MI 8 IMUs only within 60 s. (**a**) IMU dead-reckoning horizontal errors within 60 s at Outage A. (**b**) Google Earth view of IMU mechanization trajectories at Outage A. (**c**) IMU dead-reckoning horizontal errors within 60 s at Outage B. (**d**) Google Earth view of IMU mechanization trajectories at Outage B.

The results show diverging tracks from the reference solution in the Google satellite views, and quantitatively demonstrated by strictly increasing horizontal errors with time. The time series and table show higher horizontal errors from Mi 8 (blue) than from Xsens (red) for the complete duration of Outage A and after 20 s for Outage B. The Xsens has a slower diverging rate, suggesting a less noisy raw IMU performance. In general, the result suggests robust ultra-low-cost IMU performance for an outage less than 10 s comparable to that of automotive-grade IMUs such as the Xsens.

Another key observation is that the IMU-only solution shows better accuracy through Outage B compared to during Outage A. Theoretically, IMU error propagation is complicated and influenced by a range of factors including initialization errors, IMU timing discrepancy, finite iteration rates, and noises. Measurement biases, mainly from gyroscope also play a critical role in impacting IMU dead-reckoning solutions, and the positioning errors $\delta X$ in the respective axis can be represented as Equation (4) [42]:

$$
\begin{aligned}
\delta X_a &\approx \frac{1}{2} b_a t^2 \\
\delta X_g &\approx \frac{1}{6} b_g g t^3
\end{aligned}
\tag{4}
$$

where $\delta X_a$ and $\delta X_g$ are the position errors caused by accelerometer and gyroscope biases ($b_a$ and $b_g$), respectively; $g$ is gravity and $t$ is the IMU mechanization time. In other words, a $10^{-4}$ m/s$^2$ accelerometer bias and a $10^{-3}$ rad/s gyro bias theoretically produce position errors of 0.5 cm and 1.6 m, respectively, in 10 s.

Further analysis of these IMU biases is presented in Figure 6. The experiments were conducted in kinematic environments that started with a straight path. The initial biases are set as zero and it takes time for both of IMUs' biases to converge. The engine has run for 5 min prior to Outage A compared to 15 min prior to Outage B. From Figure 6, it can be observed that the bias states are better converged at the 15-min mark than the 5-min mark, potentially giving a better bias estimation. In addition, the Xsens IMU has more stable bias estimation which converges faster than for the smartphone IMU. For the accelerometers, the Mi 8 IMU does not show a clear converging pattern. For the gyroscopes, the Mi 8 IMU takes almost 10 min more to reach a stable state compared to the Xsens. This comparative performance is expected owing to lower-cost and performance of the smartphone IMU. As a consequence of these two reasons, the early unconverged estimated biases lead to the lower dead-reckoning performance at Outage A compared to Outage B.

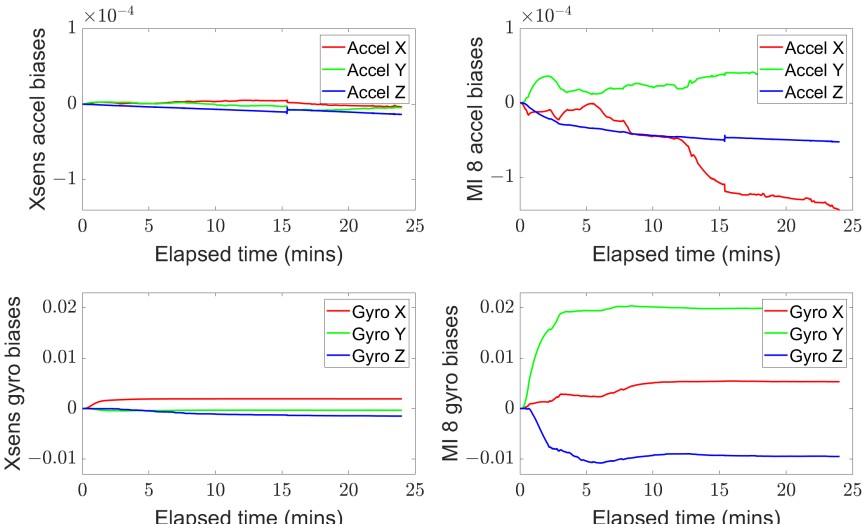

**Figure 6.** Times series of estimated IMU accelerometer and gyroscope biases with along three axes.

### 4.2. Kinematic Performance Assessment with Different Processing Strategies

Following the demonstration of IMU-only performance induced outages, this study returns to different combinations of PPP-IC smartphone processing and fusion with native IMU. In assessing the benefits of including more observations, the accuracy of three different combinations of GNSS signals processing, namely the single-frequency PPP (SF-PPP), dual-frequency PPP (DF-PPP), as well as single- and dual-frequency PPP (SFDF-PPP) are evaluated based on the time series using one of the five datasets (30 November 2021—Mi 8). All datasets are summarised afterwards to broaden the conclusions. It is important to note that GPS + GLONASS + Galileo + BeiDou (GREC) four constellations are processed both in SF-PPP and SFDF-PPP combinations, but the latter also considers the GPS L5 and Galileo E5a signals, which captures the benefits of dual-frequency PPP (GE).

#### 4.2.1. Single-Frequency PPP-IC/IMU Processing

Figure 7a illustrates the horizontal error (with respect to the reference solution) comparison for SF-PPP (GREC) processing. To better appreciate the impact of the IMU measurements, the positioning solutions processed with PPP-only, tightly-coupled PPP/MI8 IMU, and PPP/Xsens IMU integration are represented as red, blue, and green lines, respectively, and the EKF horizontal position precision estimates are included in comparison to the real horizontal position errors with the same time series. The two black rectangles mark the two aforementioned outages (A and B). As expected, the horizontal errors relying on GNSS alone for GNSS PPP (red line) increase sharply when the vehicle passes through these sky obstructions. Correspondingly, it can be observed from Figure 7b that the horizontal precision estimated by the EKF increases significantly at elapsed times 5.5 and 15.4 min, proving that the EKF estimated precision is sensitive to the GNSS outages. However, the magnitude of the precision level from the outages was significantly higher compared to the actual errors, reflecting that the estimated precision cannot serve as the representative of actual position error.

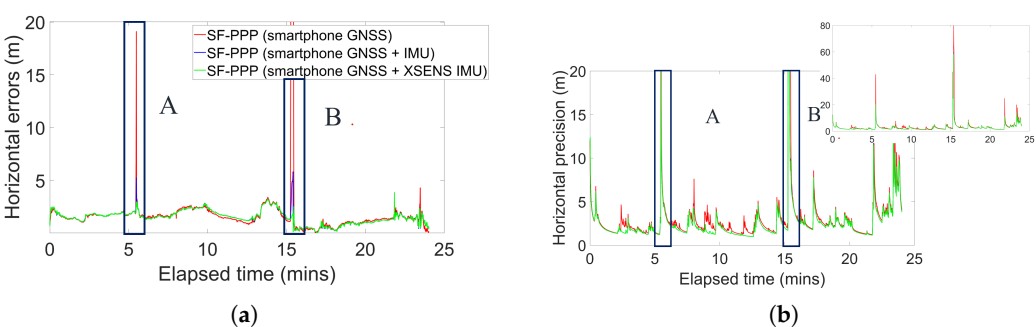

**Figure 7.** Horizontal error with respect to the reference trajectory and filter estimated precision comparison for single-frequency PPP processing using smartphone GNSS measurements, GNSS + IMU measurements, and GNSS and Xsens IMU measurements. (**a**) Horizontal position error. (**b**) Horizontal precision estimate.

In contrast, the tightly-coupled PPP/IMU processing approach significantly mitigates the positioning errors during the outages, especially with the aid of Xsens IMU and to a lesser extent with the smartphone IMU. In addition, 4% improvements from estimated precision can be observed as well when integrating IMUs. Table 3 provides the horizontal error summary statistics from the different sensor fusion combinations. The horizontal position standard deviation (about the mean) error statistics shows the repeatability of the estimates, which are reduced from 6.0 m with just GNSS PPP measurements to under 1 m when either IMU's measurements are incorporated. Importantly, the rms (i.e., the accuracy) of SF-PPP solution is improved from 6.3 m to 1.7 m and 1.6 m by integrating the smartphone and Xsens IMU, respectively. The IMU provides little gain for 95th percentile error as the tightly-coupled PPP/IMU solutions depend heavily on GNSS in open-sky

environments, implying the fusion strategy did no harm to the PPP solutions when visible satellites are sufficient. However, the IMU significantly constrains the maximum error (100 percentile) when GNSS measurement limitations are most sensitivity to position estimation. In addition, Table 4 quantifies the error growth with respect to the outage time during Outage B. Owing to insufficient satellites, the horizontal positioning performance of traditional SF-PPP degrades significantly and the error exceeds 100 m when the elapsed time is 925 s. In contrast, maximum horizontal errors of 5.8 m and 2.4 m can be observed when the vehicle passed through Outage B with the aid of smartphone and Xsens IMUs, respectively. Refs. [25,26] show similar smartphone SF-PPP positioning performance through kinematic vehicle experiments, but solutions are improved significantly by integrating IMUs in this study.

**Table 3.** Horizontal errors with different sensor fusion combinations for single-frequency PPP processing.

| Processing Strategy | Std Dev (m) | Rms (m) | 95th Percentile Error (m) | Maximum Error (m) |
|---|---|---|---|---|
| SF-PPP | 6.0 | 6.3 | 2.7 | 104.1 |
| SF-PPP/IMU (MI8) | 0.7 | 1.7 | 2.6 | 5.8 |
| SF-PPP/IMU (Xsens) | 0.6 | 1.6 | 2.6 | 3.8 |

**Table 4.** Horizontal errors growth with different sensor fusion combinations for single-frequency PPP over Outage B.

| Elapsed Time (s) | 917 | 918 | 919 | 920 | 921 | 922 | 923 | 924 | 925 |
|---|---|---|---|---|---|---|---|---|---|
| SF-PPP Horizontal errors (m) | 13.3 | 25.6 | 37.4 | 48.5 | 59.9 | 71.9 | 81.9 | 95.6 | 104.1 |
| SF-PPP/IMU (MI8) horizontal errors (m) | 1.8 | 2.3 | 2.9 | 3.8 | 4.5 | 4.8 | 4.9 | 5.1 | 5.8 |
| SF-PPP/IMU (Xsens) horizontal errors (m) | 1.3 | 1.4 | 1.4 | 1.5 | 1.6 | 1.7 | 1.8 | 2.0 | 2.4 |

### 4.2.2. Dual-Frequency PPP-IC/IMU Processing

Similarly, Figure 8 shows the time series of horizontal errors and EKF estimated precision when processing DF-PPP (GE) with different IMU sensors. It is anticipated that the DF-PPP performance is not as stable as SF-PPP due mainly to the limited number of tracked dual-frequency constellations and satellites. Thus, numerous fluctuations occurred not only during the two outages A and B, but also in some unexpected scenarios such as C on a highway, which may be owing to dual-frequency signal blockage by passing trucks.

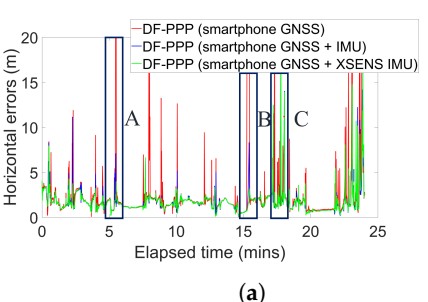

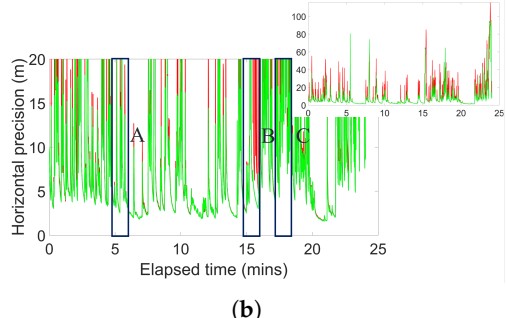

(**a**)      (**b**)

**Figure 8.** Horizontal error with respect to the reference trajectory and filter estimated precision comparison for dual-frequency PPP processing using smartphone GNSS measurements, GNSS+IMU measurements, and GNSS and Xsens IMU measurements. (**a**) Horizontal position error. (**b**) Horizontal precision estimate.

In spite of the lack of dual-frequency signals, it is somewhat surprising that with the aid of an IMU, the standard deviation of horizontal errors are improved from 6.0 m to the

sub-metre level, and the horizontal positioning error is mitigated from 8.3 m to 2.4 m and 2.3 m with MI8 and Xsens IMU, respectively (see Table 5). Similarly, Table 6 shows the horizontal error growth for dual-frequency PPP over Outage B. Furthermore, by leaving out the largest 5% of position outliers, the 95th percentile level of horizontal error indicates that the DF-PPP/IMU combination is capable of providing almost the same level of positioning accuracy as the SF-PPP/IMU solution, showing the benefits of the sensor fusion.

**Table 5.** Horizontal errors with different sensor fusion combinations for dual-frequency PPP processing.

| Processing Strategy | Std Dev (m) | Rms (m) | 95th Percentile Error (m) | Maximum Error (m) |
|---|---|---|---|---|
| DF-PPP | 7.8 | 8.3 | 6.7 | 139.2 |
| DF-PPP/IMU (MI8) | 1.6 | 2.4 | 3.3 | 23.3 |
| DF-PPP/IMU (Xsens) | 1.5 | 2.4 | 3.2 | 23.5 |

**Table 6.** Horizontal errors growth with different sensor fusion combinations for dual-frequency PPP over Outage B.

| Elapsed Time (s) | 917 | 918 | 919 | 920 | 921 | 922 | 923 | 924 | 925 |
|---|---|---|---|---|---|---|---|---|---|
| DF-PPP Horizontal errors (m) | 37.8 | 49.5 | 60.7 | 72.0 | 82.1 | 95.2 | 109.2 | 123.9 | 139.2 |
| DF-PPP/IMU (MI8) horizontal errors (m) | 1.0 | 1.4 | 1.9 | 2.5 | 3.2 | 4.1 | 5.3 | 6.7 | 8.2 |
| DF-PPP/IMU (Xsens) horizontal errors (m) | 0.9 | 1.1 | 1.1 | 1.2 | 1.3 | 1.6 | 2.0 | 2.5 | 3.1 |

To explicitly appreciate the benefits of added L5/E5a signals, Figure 9 compares the smartphone horizontal positioning performance produced from single-frequency PPP (red line) and dual-frequency PPP (blue line) processing with same dual-frequency satellites. Dual-frequency PPP outperforms single-frequency PPP, and the latter approach cannot provide stable positioning performance with accuracy lower than 20 m during the elapsed time between 15 and 20 min, owing to insufficient and inaccurate single-frequency measurements.

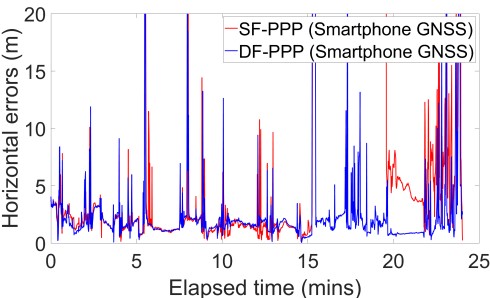

**Figure 9.** Horizontal error with respect to the reference trajectory for single-frequency PPP and dual-frequency PPP processing with same satellites.

The solution shows the kinematic DF-PPP accuracy without sensor fusion is improved by 1–2 m as per the literature in similar experimental settings [26]. The pattern of higher DF-PPP errors than SF-PPP errors due to the lack of dual-frequency signals is also found in existing literature [25]. The solution shows accuracy degradation of DF-PPP solution of about 3 m compared to similar settings with phones on top of vehicle, whereas the discrepancy in fused horizontal accuracy is within ±1 m [35].

### 4.2.3. Single- and Dual-Frequency PPP-IC/IMU Processing

Single- and dual-frequency PPP (SFDF-PPP) processing is the main focus of this study, which utilizes not only GPS L1, GLONASS L1, Galileo E1, BeiDou B1 signals, but also GPS L5 and Galileo E5a signals if they are available, as well as ionospheric constraints from

GIM. Figure 10 displays the time series of horizontal errors and precision for SFDF-PPP (GREC) processing strategies. In general, SFDF-PPP processing inherits the advantages of the aforementioned two processing strategies with more observations and good GNSS geometry, resulting in more stable positioning.

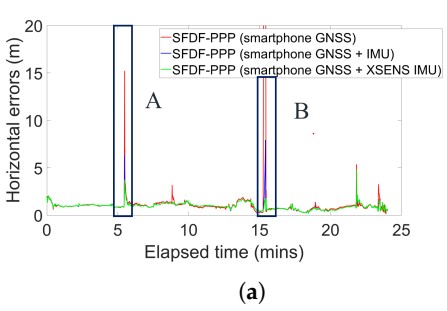
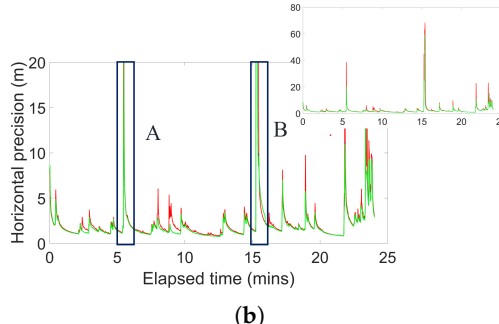

(**a**)                              (**b**)

**Figure 10.** Horizontal error with respect to the reference trajectory and filter estimated precision comparison for single- and dual-frequency PPP processing using smartphone GNSS measurements, GNSS + IMU measurements, and GNSS and Xsens IMU measurements. (**a**) Horizontal position error. (**b**) Horizontal precision estimate.

Table 7 summarises the horizontal error statistics for SFDF-PPP processing with and without IMUs. On the one hand, accessible L1 and L5/E5a signals increases smartphone positioning overall performance, namely 5.0 m and 5.2 m of horizontal error standard deviation and rms can be observed, respectively, which are the best solutions among pure GNSS PPP strategies. On the other hand, the integration of IMUs further improves positioning accuracy and resilience, especially for all scenarios that suffer from GNSS outages (outliers, see Table 8). Compared to SFDF-PPP processing, 92% and 79% improvements are observed by integrating smartphone native IMU in terms of horizontal standard deviation and rms, respectively. In addition, use of the Xsens IMU enables improved horizontal solutions. Though slight improvement can be found for 95th percentile error, the benefits of IMUs mainly contribute by mitigating the GNSS outliers and the maximum horizontal errors reduce from 90.3 m to under 10 m.

**Table 7.** Horizontal errors with different sensor fusion combinations for single- and dual-frequency PPP processing.

| Processing Strategy | Std Dev (m) | Rms (m) | 95th Percentile Error (m) | Maximum Error (m) |
|---|---|---|---|---|
| **SFDF-PPP** | 5.0 | 5.2 | 1.7 | 90.3 |
| **SFDF-PPP/IMU (MI8)** | 0.4 | 1.1 | 1.5 | 7.9 |
| **SFDF-PPP/IMU (Xsens)** | 0.3 | 1.0 | 1.5 | 4.8 |

**Table 8.** Horizontal errors growth with different sensor fusion combinations for single- and dual-frequency PPP over Outage B.

| Elapsed Time (s) | 917 | 918 | 919 | 920 | 921 | 922 | 923 | 924 | 925 |
|---|---|---|---|---|---|---|---|---|---|
| **SFDF-PPP Horizontal errors (m)** | 10.3 | 19.5 | 28.9 | 38.5 | 48.0 | 58.1 | 68.9 | 79.0 | 90.3 |
| **SFDF-PPP/IMU (MI8) horizontal errors (m)** | 0.9 | 1.3 | 1.7 | 2.2 | 2.8 | 3.6 | 4.7 | 6.2 | 7.9 |
| **SFDF-PPP/IMU (Xsens) horizontal errors (m)** | 0.8 | 0.9 | 1.1 | 1.3 | 1.5 | 1.7 | 2.0 | 2.3 | 2.7 |

### 4.3. General Kinematic Performance Statistics

Road tests are subject to a series of factors impacting positioning performance which cannot be controlled. These factors include various traffic conditions, passing of large vehicles, satellite availability, and tropospheric/ionospheric conditions at the time. Therefore,

this study summarises four more datasets based on three days of road tests, while also presenting additional performance results using a Samsung Galaxy S21+. The additional datasets are collected on identical routes (with identical buildings), but due to specific traffic levels and traffic lights they do not have identical duration. Table 9 summarising all five datasets. The collection of all five datasets illustrates the repeatability of the analyzed results and provides added confidence in the performance statistics.

**Table 9.** Summary of all road tests.

| Road Test # | Phone Model | GPS Date of Collection (2021) | Duration |
|:---:|:---:|:---:|:---:|
| **M1** | MI 8 | 8 August | 01:52–02:18 |
| **S1** | S21+ | 8 August | 01:52–02:18 |
| **M2** | MI 8 | 12 October | 00:44–01:00 |
| **S3** | S21+ | 30 November | 02:04–02:28 |
| **M3** | MI 8 | 30 November | 02:04–02:28 |

The following set of time series illustrates in Figure 11 shows the horizontal errors through each road test. Road test M3 was analyzed in Figures 7a, 8a and 10a, so its error analysis will not be repeated. Similarly, the SFDF-PPP solution (in green) shows the highest accuracy and stability by mitigating the majority of error peaks. In comparison to the S1 PPP solutions, inconsistent performance can be noticed for dual-frequency PPP-only processing for the S3 dataset (see Figure 11d) owing to insufficient dual-frequency satellites, which are reduced from, on average, 7.6 (S1) to 5.4 (S3). The inclusion of multiple datasets bring more challenges in smartphone native PPP/IMU solutions owing to the low-cost IMU hardware. Correspondingly, some smartphone PPP/IMU outliers can be observed such as jumps in M1 at elapsed time ~8 min. In spite of these dataset variations, PPP/IMU solutions have more accurate performance compared to PPP GNSS-only solutions for most epochs.

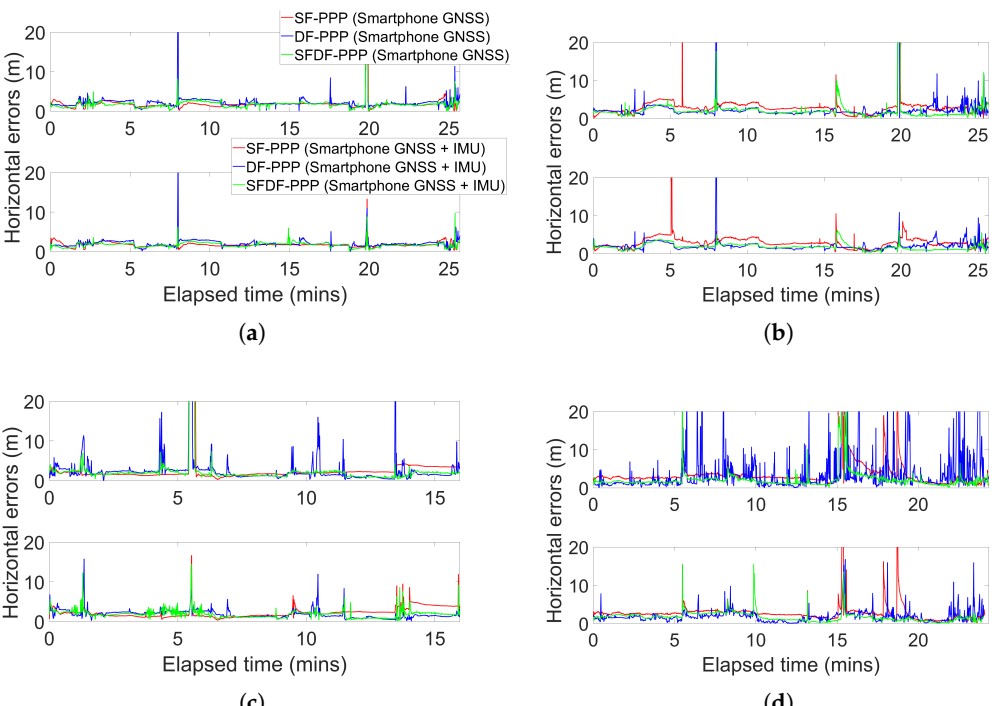

**Figure 11.** Horizontal error with respect to the reference trajectory comparison for different datasets processed by single-frequency, dual-frequency, single- and dual-frequency PPP and PPP/IMU strategies. (**a**) Horizontal position error for dataset M1. (**b**) Horizontal position error for dataset S1. (**c**) Horizontal position error for dataset M2. (**d**) Horizontal position error for dataset S3.

To generalise the findings from all collected datasets in comparing the three PPP-processing strategies and the impact of IMU fusion, the horizontal rms errors are concluded in Figure 12a and 95th percentile errors are concluded in Figure 12b. Statistically, all six combinations of the processing strategies are presented in Table 10. Figure 12a emphasizes the effect from IMU fusion through all epochs during the road tests, thus covering various error peaks and the corner cases encountered. There is a reduction of standard deviation (of horizontal errors) from 6.5 m to 1.0 m, and 6.8 m to 1.9 m of rms with SFDF-PPP processing. Similar observations can be made for SF-PPP and DF-PPP processing. The results show a potential key role of IMU fusion in generating a more stable smartphone navigation solution in real-world scenarios. For higher-cost GNSS receiver/IMU combinations, the result is well expected. However, as the results are produced from ultra-low-cost smartphone hardware, this results demonstrate the sense of necessity of sensor fusion to achieve more demanding user requirements with given cost constraints.

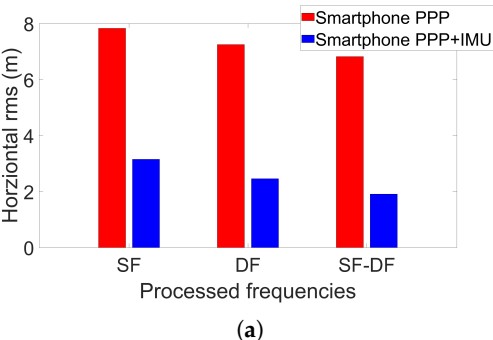 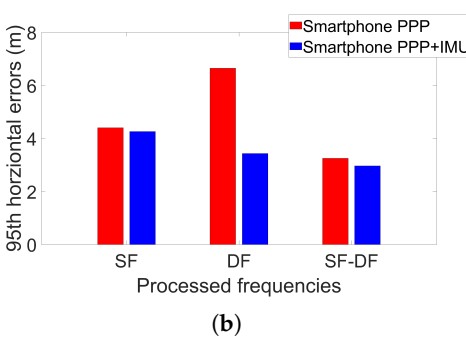

**Figure 12.** Overall horizontal errors from different processing strategies. (**a**) Overall horizontal rms comparison. (**b**) 95th percentile horizontal errors comparison.

**Table 10.** Overall statistics for different processing strategies.

| Processing Strategy | Std Dev (m) | Rms (m) | 95th Percentile Error | Maximum Error (m) |
|---|---|---|---|---|
| SF-PPP | 7.3 | 7.8 | 4.4 | 175.5 |
| SF-PPP/IMU | 2.2 | 3.2 | 4.3 | 70.6 |
| DF-PPP | 6.6 | 7.3 | 6.6 | 139.2 |
| DF-PPP/IMU | 1.5 | 2.5 | 3.4 | 57.5 |
| SFDF-PPP | 6.5 | 6.8 | 3.2 | 138.6 |
| SFDF-PPP/IMU | 1.0 | 1.9 | 2.9 | 19.9 |

Figure 12b emphasizes the majority case for all road tests, showing 95th percentile horizontal error. As in Figure 12a, the fusion with the native IMU shows improvement over PPP-only solutions, despite being less significant in SF and SFDF-PPP processing. As expected, without IMU fusion, DF-PPP displays lower solution quality compared to SF-PPP and SFDF-PPP due to inconsistencies in tracking dual-frequency satellites from the vehicle dashboard. However, by fusing GNSS measurements with IMU measurements, DF-PPP/IMU combination shows a reduction of 95th percentile error from 4.3 m to 3.4 m compared to SF-PPP/IMU in contrast to PPP-only processing where there is an increase from 4.4 m to 6.6 m, indicating that though DF-PPP on smartphones suffer from inconsistent measurements, it can still provide better solutions than SF-PPP when inertial sensors are used. The SFDF-PPP strategy shows the most robust accuracy among all three PPP processing strategies. In addition, compared to the existing literature, such as [28] which achieved approximately 0.85 m horizontal rms in open-sky walking environments, the smartphone native SFDF-PPP/IMU solutions are capable of providing comparable positioning performance in real-world driving scenarios, showing great potential for vehicle and other smartphone-based applications.

### 4.4. Correlation Analysis of Single- and Dual-Frequency PPP Horizontal Accuracy under Different Environments

The previous results have highlighted the importance of PPP processing with SFDF observations. To better appreciate the benefit of this combination strategy, metrics such as the number of satellites and observations, as well as PDOP (position dilution of precision) are compared for the three chosen PPP processing strategies. The kinematic dataset M3 collected on 30 November 2021 (DOY 331) with the MI 8 is used for the analysis (see Figure 13).

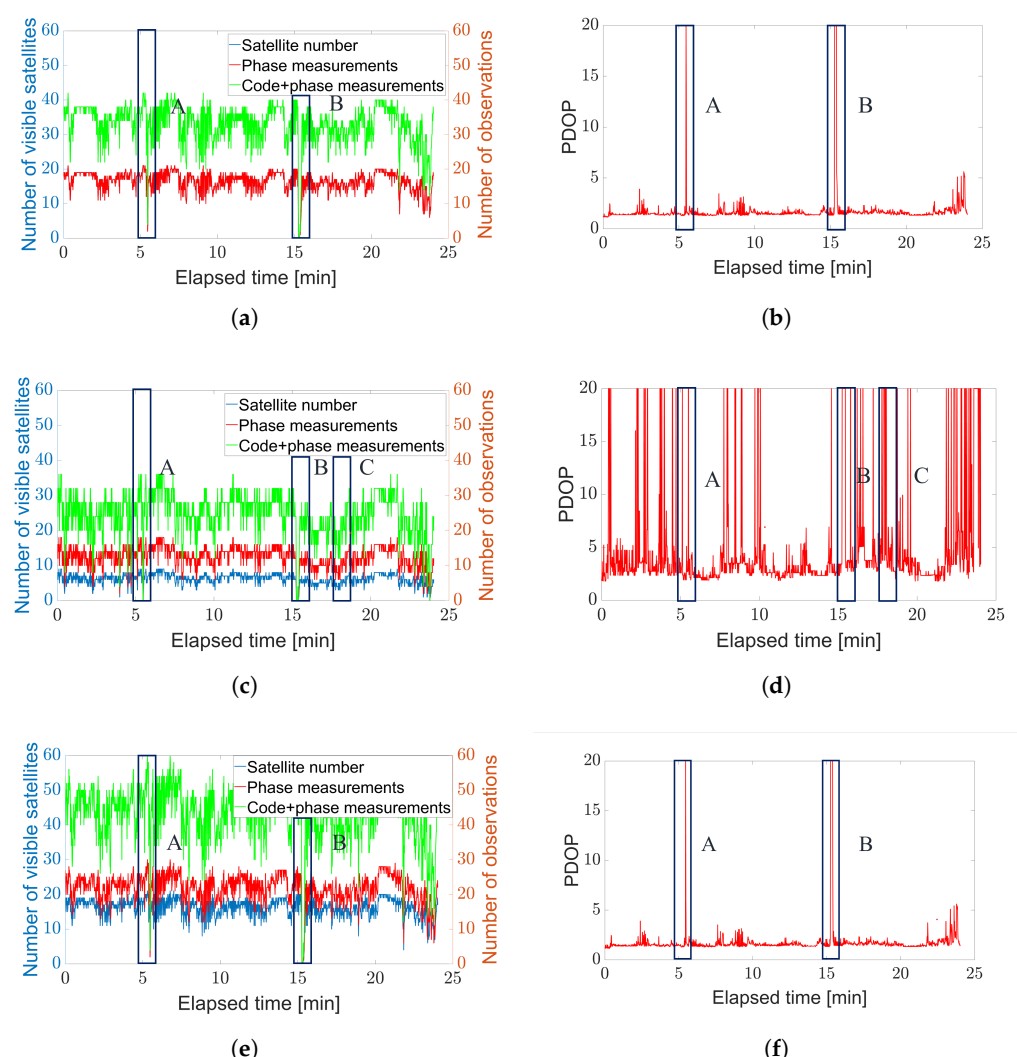

**Figure 13.** Time series of the number of satellites, observations, and PDOP in three processing strategies. (**a**) Time series of the number of satellites and observations in SF processing. (**b**) Time series of PDOP in SF processing. (**c**) Time series of the number of satellites and observations in DF processing. (**d**) Time series of PDOP in DF processing. (**e**) Time series of the number of satellites and observations in SFDF processing. (**f**) Time series of PDOP in SFDF processing.

As illustrated in Figure 13, it is expected that the number of processed satellites of SF (16.6), and SFDF (16.1) strategies far outweigh the DF (6.1) processing. Accordingly, the SF-PPP PDOP (1.7) is almost identical to the PDOP for SFDF-PPP (1.7), as opposed to the DF-PPP PDOP (4.7) owing to the limited number of tracked satellites. The SFDF method, nevertheless, is able to track an average of 43.6 observations including phase and code measurements, which is significantly larger than the other two approaches, leading to better positioning performance. From Figure 13c,d, the number of observations and PDOP

fluctuate significantly not only at the outages A and B, but in some scenarios where the dual-frequency satellites are not sufficient owing to buildings or trucks blockage, such as C (marked by the black rectangle), indicating that the DF-PPP is more sensitive and vulnerable to multipath effects and signal outages, particularly in challenging environments.

The impact on horizontal errors from the number of observations and PDOP can be seen in Figure 14. Figure 14a shows the correlation between horizontal errors and the number of observations based on all datasets for SFDF-PPP processing, and by increasing the number of observations that the filter can draw on, a negative correlation (−0.37) is observed with horizontal positioning accuracy. However, excessive observations (more than 40) only provide marginal improvement on positioning, due mainly to the limitation of satellite geometry. Meanwhile, a positive but relatively low correlation coefficient of 0.28 is observed between PDOP and positioning accuracy in Figure 14b, which suggests PDOP may not serve as an ideal indicator for smartphone PPP performance. It is worth mentioning that the scenarios where insufficient or no satellites observed are excluded from this correlation analysis.

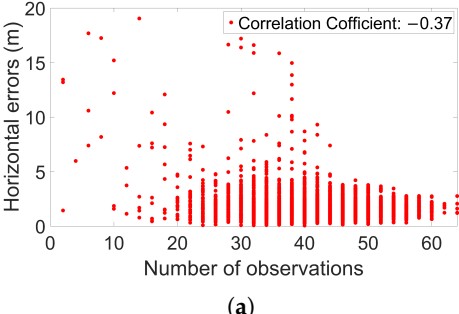
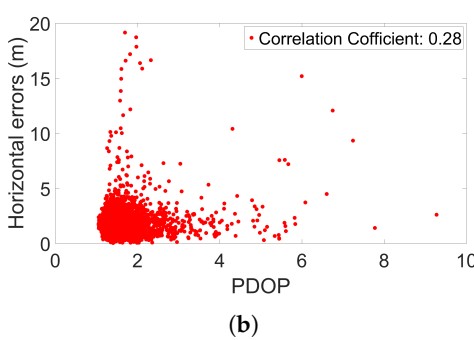

(**a**)        (**b**)

**Figure 14.** Correlation of the horizontal errors on the mean number of observations and PDOP. (**a**) Correlation of the horizontal errors on the number of observations. (**b**) Correlation of the horizontal errors on PDOP.

Table 11 compares the pre-fit and post-fit residuals for L1/E1/B1 (GREC) and L5/E5a (GE) signals in SFDF-PPP processing strategy and interestingly, both the pre-fit pseudorange and carrier-phase residuals exceed 10 m. The L5/E5a residuals are significantly lower than for L1/E1/B1, namely 3.7 cm and 2.7 cm of post-fit carrier-phase residual rms can be observed for L1/E1/B1 and L5/E5a signals, respectively. The aforementioned correlation and residual rms analysis provide insight into how SFDF-PPP solutions benefit from the extra observations with more accurate L5/E5a frequencies.

**Table 11.** Pre-fit and post-fit residual rms for single- and dual-frequency PPP processing.

| Frequency | L1/E1/B1 (GREC) | | L5/E5a (GE) | |
|---|---|---|---|---|
| Measurement | Pseudorange | Carrier-Phase | Pseudorange | Carrier-Phase |
| Pre-fit residual rms (m) | 11.7 | 11.2 | 10.3 | 10.1 |
| Post-fit residual rms (m) | 3.5 | 0.037 | 0.8 | 0.027 |

To explore the reason behind the unexpected magnitude of pre-fit residual rms, Figure 15 illustrates the time series of estimated GPS clock offset and clock difference of two adjacent epochs. Some fluctuations in smartphone estimated GPS receiver clock offset are due to their very low-cost oscillators compared to, e.g., geodetic GNSS receivers embedded with voltage-control or more stable oscillators. Such unstable estimated clock offsets lead to the same variations of their clock difference. In other words, the relatively large rms of pre-fit residuals is caused by the inaccurate EKF prediction utilizing the estimated receiver clock offset from the previous epoch, which cannot reflect the current or true estimated clock offset. These differences range from −8 m to 20 m, as seen in Figure 15b.

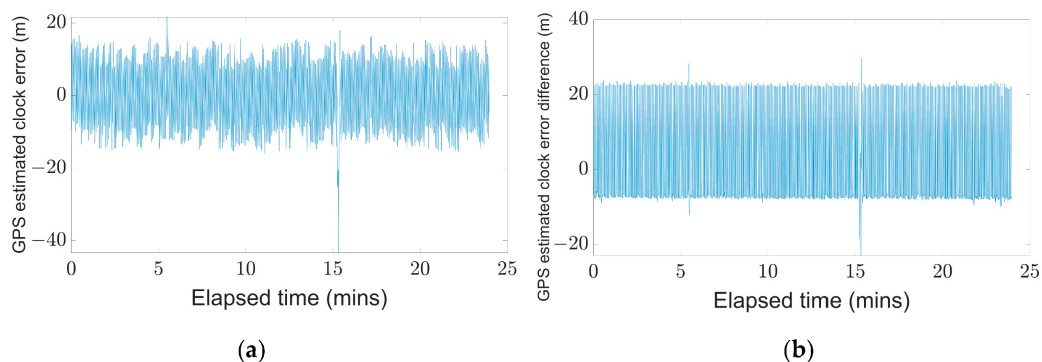

**Figure 15.** Time series of estimated GPS clock offset and corresponding clock difference. (**a**) Estimated GPS clock offset. (**b**) Estimated GPS clock difference of adjacent epochs.

To specify the benefits of incorporating smartphone IMU measurements under different environments, Figure 16 presents the horizontal rms with the number of satellites used in SFDF-PPP processing. On account of buildings and tunnels limiting portions of the sky, the minimum 4 satellites are not always available for smartphone GNSS positioning in suburban driving. Observing both Table 12 and Figure 16, the smartphone IMU can bridge these positioning gaps and significantly mitigate positioning errors when visible satellites are insufficient, e.g., a significant level of 98% and 87% horizontal positioning improvements is observed with the aid of the smartphone IMU measurements when the number of satellites are reduced to 4 and 5, respectively. The smartphone PPP/IMU solution is capable of providing stable and continuous positioning solutions, and the sensor fusion solution maintains the same grade of accuracy performance even with fewer satellites. The possible reason is that smartphone PPP/IMU solutions not only rely on GNSS satellite measurements, but also are relevant to predicted state covariance, previous estimated states, and IMU raw measurement quality.

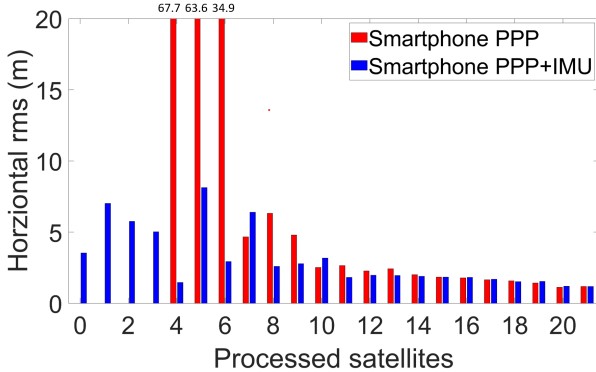

**Figure 16.** Correlation of the horizontal errors on the number of satellites.

**Table 12.** Statistics of horizontal errors with processed satellites for single- and dual-frequency processing.

| Processed Satellites per Processing Strategy | 0 | 1 | 2 | 3 | 4 | 5 | 6 | 7 | 8 | 9 | >9 |
|---|---|---|---|---|---|---|---|---|---|---|---|
| SFDF-PPP | - | - | - | - | 67.7 | 63.6 | 34.9 | 4.6 | 6.3 | 4.7 | 1.7 |
| SFDF-PPP/IMU | 3.5 | 7.0 | 5.7 | 5.0 | 1.5 | 8.1 | 2.9 | 6.4 | 2.6 | 2.7 | 1.6 |

## 5. Conclusions and Future Work

This paper presents several noteworthy contributions by integrating single- and dual-frequency GNSS observations with inertial measurements from native smartphone sensors. A total of five datasets were collected from three days of road tests in suburban driving environments were analyzed and based on these investigations, this paper addresses the posed questions:

1. *How does single- and dual-frequency PPP processing improve smartphone GNSS positioning performance and how does it compare with other PPP processing strategies (single-frequency PPP and dual-frequency PPP) in GNSS challenged environments?* Verified by kinematic experiments, single- and dual-frequency PPP (SFDF-PPP) processing captures the essence of single-frequency and dual-frequency PPP processing strategies as all single-frequency signals (GREC), as well as dual-frequency signals including GPS L5 and Galileo E5a are utilized. SFDF-PPP processing with an average of 43.6 observations from all GREC constellations lead to an overall 6.8 m horizontal rms and 3.2 m of 95th percentile horizontal error, which outperforms other PPP strategies.

2. *How does smartphone IMU dead-reckoning perform compared to other low-cost MEMS IMU? And how does the inclusion of the smartphone inertial sensor affect PPP solutions?* Through the dead-reckoning experiments, it was found that the smartphone IMU is capable of providing metre-level positioning accuracy for an outage of less than 10 s with good bias estimation, which is comparable to that of the more capable Xsens IMU. Furthermore, the inclusion of the smartphone inertial sensor can reduce GNSS outliers and provide continuous and accurate navigation solutions in GNSS challenged environments. In addition, significant levels of 59%, 66%, and 72% improvement in overall horizontal rms can be observed from single-frequency PPP, dual-frequency PPP, as well as single- and dual-frequency PPP, respectively.

3. *What is the "best" positioning performance that smartphones can achieve with multi-GNSS PPP/IMU integration in real-world driving environments?* Based on these investigations, single- and dual-frequency PPP/IMU integration is an optimal solution among other strategies for smartphone positioning. The general suburban performance statistics show that smartphones can achieve 1.9 m overall horizontal rms with 1.0 m standard deviation in real-world driving scenarios with the consideration of multiple multipath profiles, which have not been seen in previous studies.

Future work will utilize additional holonomic and non-holonomic constraints for vehicle motion and explore the impacts of vehicle-smartphone relative motion when drivers move the phone. Furthermore, this work will be extended to more complex and obstructed environments with an adaptive EKF and fusion with additional sensors.

**Author Contributions:** Conceptualization, D.Y., S.Y. and S.B.; Methodology, D.Y. and S.Y.; Software, D.Y.; Validation, D.Y. and S.Y.; Writing—Original Draft Preparation, D.Y. and S.Y.; Writing—Review and Editing, S.B.; Supervision, S.B. All authors have read and agreed to the published version of the manuscript.

**Funding:** Funding was provided by the Natural Sciences and Engineering Research Council of Canada (NSERC).

**Data Availability Statement:** The data presented in this study are available from the corresponding author on reasonable request.

**Acknowledgments:** The authors would like to thank their colleagues at the GNSS Laboratory for support in collecting field data and discussing technical aspects of this work. In addition, the authors would like to acknowledge the data contribution from the International GNSS Service (IGS), German Research Centre for Geosciences (GFZ), and Centre National d'Etudes Spatiales (CNES).

**Conflicts of Interest:** The authors declare no conflict of interest.

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
