# Peer review of "Native Smartphone Single- and Dual-Frequency GNSS-PPP/IMU Solution in Real-World Driving Scenarios"

_remotesensing, doi:10.3390/rs14143286_

Round 1
Reviewer 1 Report
The paper presents interesting results in the field of smartphone PPP. The methodology is sound in general, but some aspects call for clarification:
1. The paper makes the assumption that the smartphone lies flat on the dashboard. Is the EKF applying nonholonomic constraints for the vehicle motion? Moreover, how would the results change if the user would be holding the phone in the hand (and moving it) instead?
2. Table 1 quotes constant standard deviations for the measurements AND a weighting scheme. It would seem necessary to show the standard deviation range after weighting for the processed SNR range (either by giving a numerical interval or plotting). It is stated that the std values were obtained in postprocessing. Was a very short baseline experiment attempted to measure the standard deviations directly?
3. The elevation cutoff was 10 deg. How many satellites did this discard on average?
4. Three IMUs under test are assessed in the paper (Mi, S21, and Xsens IMUs) but no information about their nominal performance is given. For the smartphones, the IMU chip models are probably not known. In that case, the most important parameters should be computed using the Allan variance. This is important as it characterizes the quality of the sensors that yielded the performance quoted in the article.
5. In Fig. 6, the accelerometer bias y-axis scale seems very small. Presumably, the initial bias values were removed to fit both lines in the same axes. The authors could consider plotting the phone biases in one plot (all three axes) and Xsens to another, which would probably allow for including the absolute biases.
6. Fig 7: the subplots are impossible to compare because the y-axis scales are so different. The spikes at the outages could probably be cropped without losing important information, which would allow using the same y-axis scale.
7. The experiments include gaps, but their duration seems to be too short to affect the 95th percentiles; only the max errors now characterize the performance under outages. Quantifying the error growth as a function of outage time would give a more generic result.
8. Section 4.2.2: did the authors attempt running SF-PPP/IMU with only those satellites that were available to DF-PPP/IMU? This would help quantify the true performance gain of dual-frequency processing.
9. Fig. 10d: the DF-PPP for S3 looks very noisy in comparison with the other runs, which probably also affects Fig. 11. Was a reason for degraded DF-PPP with S3 identified?
10. Fig. 12: plotting the total amount of observations on top of the number of satellites doesn't seem to add much information; in fact, since it combines measurement types of very different characteristics (code and phase), it is prone to be misleading. The authors could consider plotting the number of code and phase measurements separately (or alternatively, phase measurements and total number of measurements if it is easier to plot) to be more informative.
11. In principle, the correlation of PDOP and positioning accuracy can be expected, but still I am a bit surprised how good the correlation is. Does the result suggest that the underlying assumption of independent and identically distributed measurement errors is actually still quite valid?
12. Table 12: the pre-fit residuals for code and carrier measurements are practically identical, and the paper claims that this is due to multipath and noise. This calls for further evidence: if the measurements were indeed noisy, then the post-fit residuals should remain large (because the noise should be filtered out), right? To me it sounds like the EKF cannot predict the state correctly (e.g. very unstable local oscillators causing difficulties predicting the clock bias). Please clarify.
Reviewer 2 Report
This paper presents a sensor fusion technique using Precise Point Positioning (PPP) and the inertial sensors in smartphones, combined with a single- and dual-frequency (SFDF) optimisation scheme for smartphones. After carefully reading, I find that this paper is extremely interesting. But there are still some problems in this paper.
1. State of the art is largely incomplete, many recent works dealt with sensor fusion technique combined with a single- and dual-frequency (SFDF) optimisation are not even mentioned.
2. Novelty of the contribution is not clear (and in my opinion very limited): proposal seems to be a simple concatenation of known techniques with not theoretical support, such as Precise Point Positioning and single- and dual-frequency (SFDF) optimisation scheme.
3. Many Figures have poor visibility and inconsistent sizes. It is suggested that the author redraw the Figures using different colors, such as Figures 1 and 2.
4. English style should be improved and many concepts are hard to follow and understand.
5. The conclusions and future work are too long, and the authors need to refine it further. It is suggested that only important innovations and experimental conclusions should be retained.
Round 2
Reviewer 2 Report
Thank the authors for their efforts. The authors have adequately addressed all my concerns in the review, and did a good job to revise and improve the paper. The paper now is suitable for publication in Remote Sensing in its current form.
Author Response
Thanks so much for reviewing our paper!